

**Mars MOURA magnetometer demonstration for high resolution mapping on**
**terrestrial analogues**
Marina Diaz-Michelena[1], Rolf Kilian[2,3], Ruy Sanz[1], Francisco Rios[2], Oscar Baeza[2]
[1] Payloads and Space Sciences Department, INTA, Ctra. Torrejón – Ajalvir km 4, 28850 Torrejón de
Ardoz, Spain
[2] Geology Department, University of Trier, Behringstrasse, 54286 Trier, Germany
[3] University of Magellanes, Punta Arenas, Chile

**Abstract**
Satellite-based magnetic measurements of Mars indicate complex and very strong
magnetic anomalies, which led to an intensive and long-lasting discussion about their
possible origin. To investigate the origin of these anomalies MOURA vector
magnetometer was developed for future on ground surveys on Mars and tested on
various terrestrial analogues characterised by most distinct magnetic anomalies of
their basement rocks: (1) A magnetite body of EL Laco (up to +110,000 nT) and its
transition to surrounding andesites (< +2,000 nT) in the Northern Andes of Chile
showing the highest local magnetic anomalies. The magnetite-bearing ore body has
highly variable local anomalies due to their complex formation history where a
significant dispersion in paleo-orientations has been previously reported, while our
vector data show relatively uniform and probably induced declinations. (2) A basaltic
spatter cone of the Pali Aike volcanic field, in Southern Chile, was characterised by very
strong magnetic anomalies along the crater rim (up to +12,000 nT), controlled by the
amount of single domain magnetites in the ground mass of the basalts. Due to their
strong remanent signature paleodeclinations of the lavas and reorientations of
collapsed blocks could be constrained by the vector data. (3) The Monturaqui
meteorite crater (350 m diameter), in Northern Chile, shows significant variations of its
anomalies (from -2,000 to > + 6,000 nT) in restricted areas of several square metres
along its crater rim related to unexposed iron-bearing fragments of the impactor while
its granitic and ignimbritic target rocks exhibit only very weak anomalies. (4) An area
with several amphibolitic dykes which cross-cut a Cretaceous granitoid in the
southernmost Andes, where a decimetre-scale mapping was performed. In this case,
pyrrhotite is the only magnetic carrier. It was formed during hydrothermal processes
within the dykes. Very low (+40 to +120 nT) positive magnetic anomalies clearly depict
the amount of 1 to 4 vol.% pyrrhotite in these dykes, which is important as a
mineralogical indicator as well as to detect associated gold and copper enrichment.
**Keywords:** Andes, El Laco, Monturaqui, Pali Aike, Patagonian Batholith, continental
crust, martian crust, granite, basalt, mafic dykes, volcanic crater, magnetic anomalies,
magnetic carrier, magnetic susceptibility, ground magnetic survey, caesium
magnetometer, MOURA magnetometer, Koenigsberger ratios, plutonic rocks, impact
crater, iron meteorite, pyrrhotite, magnetite, hydrothermal mineralization, gold,
copper.



## 1. Introduction

Mars magnetic field has been exhaustively measured between 100 and 440 km altitude by Mars Global Surveyor (Acuña et al., 1998; Connerney et al., 2005; Morschhauser et al., 2014). These data show that the magnetic anomalies of the martian crust are up to 20 times higher than those of the Earth (Scott and Fuller, 2004). However, a profound understanding of the magnetic signature of the martian crust would require mapping at different altitudes and consequently with a different magnetic zoom apart from further petrological analyses. Since Mars presents a very low dense atmosphere, aeromagnetic surveys are not achievable in the short term. Thus, on ground magnetometry with landers and rovers seems to be the most immediate feasible technology to complement the satellite measurements. MOURA magnetometer was developed by INTA in the context of MetNet Precursor Mission to perform vector magnetometry and gradiometry during on ground prior to rover-based surveys on extra-terrestrial planets, like Mars. The instrument has a very low mass (72 g), a scalable range, high precision, low detectable fields and noise, and is capable to work in the very hard environmental conditions of Mars. Diaz-Michelena et al. (2015) describes MOURA main technical details as well as the calibration.

The first objective of the present work is to demonstrate the capability of the miniaturized MOURA instrument in a real context of terrestrial analogues by means of the inter-comparison with the data of a scalar caesium reference magnetometer (Diaz-Michelena and Kilian 2013). To do this, four different sites with a wide variability in the intensity of their magnetic signatures have been selected. A second objective is the magnetic investigation of these sites and their implication as terrestrial analogues of Mars. A further objective is the potential of high resolution mapping to show the lowest magnetic contrasts in the terrain and their correlation with the distinct magnetic carriers responsible for their signatures (Acuña et al., 1998; Connerney et al., 2005; Lillis et al., 2013).

Our selected sites include magnetic anomalies from complex geological environments where e.g. noise factors, geometrical characteristics of non-exposed rock units and effects of the terrain relief can obscure partly the interpretations concerning the kind and magnetic effects of non-exposed rock units. Thus a modelling of the magnetic anomalies is desirable in general to improve the interpretation of geometrical and compositional effects of non-exposed rocks (e.g. Eppelbaum et al., 2015; Eppelbaum and Mishne, 2011; Ialongo et al., 2014). Since we are aware of this problematics, primarily we focus on the interpretation of the effects of distinct types and compositions of exposed rocks concerning the observed magnetic anomalies. Future more detailed studies of the investigated sites should include the above mentioned magnetic modelling which is out of the scope of this magnetometer demonstration.

## 2. Methodology

### 2.1. *Magnetic Instrumentation*

Two different magnetometers have been used for the present surveys: a conventional



caesium scalar magnetometer: model G-858 MagMapper by Geometrics and MOURA
vector magnetometer designed and developed by INTA MetNet team for Mars
exploration.
G-858 is taken as the reference magnetometer because it is a well-established
hand-held instrument (8 - 9 kg) for magnetic surveys (ordnance, archaeology,
environmental, mineralogy and petroleum prospection). It has 8 hours autonomy,
provides a suitable contrast related to magnetic anomalies (8 pT/√Hz) and good
stability covering the range of the Earth magnetic field with a dynamic range between
20,000 to 100,000 nT. It also has several modes of operation: continuous and discrete
to allow users to plan the prospections grids. In contrast to MOURA, this
magnetometer only provides the intensity of the total magnetic field, its performance
is dependent on the orientation of the head respect to the field, which changes
significantly in the latitude range of the present survey, and it is restricted to areas
with gradients higher than 20,000 nT/m (Table 1).
MOURA is a vector magnetometer with two 3-axes magnetic sensors of
Anisotropic MagnetoResistance (AMR) by Honeywell to build up a compact and
miniaturized instrument (72 g mass and 67.5 $cm^3$) for Mars exploration. The power
consumption is limited to 400 - 430 mW, so it can operate during more than 10 hours
with commercial batteries, and insignificant increase of weight to the user (3 x 25 g).
The instrument is also designed for continuous or discrete modes of operation, it can
work in every orientation with no incidence in the performance, and it is practically
immune to gradients due to the small size of the transducer (μm-size).
The characteristics are designed for Mars surface environment (-90 to +20 ºC in
operation, -120 to +125 ºC in storage, and a total irradiance dose of 15 krad/s). The
resolution is limited by the transducer (0.2 nT) and the range is adapted to that of the
Earth geomagnetic field ± 65,000 nT with an extended range in the auto-offset
compensation mode of ± 130,000 nT (Table 1, Diaz-Michelena et al., 2015).
*2.2    Track performance*
The tracks have been defined to cover most of the relevant geological features of the
selected areas. Continuous and discrete modes have been selected depending on the
characteristics and heterogeneity of the sites. For example, the continuous mode has
been applied in extended areas in order to have more flexibility and speed to move. In
areas with very small-scale heterogeneities the discrete mode has been preferred. In
these cases between 5 and 7 measurements have been taken and averaged per point.
An advantage of this mode is that it can be measured directly on ground or at a fixed
distance above ground.
The positions of measuring points and tracks have been georeferenced by a
Garmin 62s GPS. The GPS tracks have been used to derive the orientation. For some
relatively small mapping areas, like Bahía Glaciares (Site 4; Fig. 1), a grid of 20 x 20 m
was previously defined by a tape measure since the error of GPS positions could be in
the order of several metres. In these cases, the lines have been used for the
orientation.
It has to be taken into account that the different data have been obtained from
multiple instruments individually without an automatic synchronism. Therefore, all the
acquisition units have been manually synchronised, and the sequence of measurement
has been done systematically as follows:
1) Marking selected measurement point.
2) GPS measurement with time stamp (with > 6 satellites at direct sight).
3) Removal of the GPS from measurement point to avoid magnetic contamination by
this device.
4) Magnetic measurement with time stamp (for G-858).
5) Magnetometer (three axes), accelerometer (three axes), temperature
measurement with time stamp (for MOURA).
The data files have been pre-processed manually (preliminary corrections of the
GPS data with the above-described information). An ad-hoc software has been
performed to include the temperature and tilt angles correction of MOURA data, to
subtract the Earth geomagnetic field with respect to total intensities in the case of G-
858 and each vector components in the case of MOURA, and to plot the different
magnitudes of the processed data.
Local magnetic field anomalies have been calculated with respect to the
International Geomagnetic Reference Field (IGFR) averaged for the month of the
surveys. At single sites the surveys were performed during less than 2 hours for which
global magnetic field data from a base station and/or from the next magnetic
observatories with minute-resolution have been considered for reference (Argentine
Islands near Antarctic Peninsula, Port Stanley and Easter Island and Huancayo ~1400
km). However, since temporal variations during all survey intervals of single sites were
less than ±10 nT (quiet days), site specific data corrections have not been applied.
**2.3    *Selected sites: mineralogical and geological context***
The selected test sites for the on ground survey are situated in or near the Southern
Andes between latitudes 20°S to 52°S (Fig. 1).
Required general site characteristics are that a) exposed rocks are relatively
unaltered, b) high resolution grids with scales of metres to centimetres can be
performed, and c) exposed rocks are representative for a large number of martian
surface rocks.
SITE 1 "El Laco". Four large magnetite bodies with a total estimated ore resource of
500 million tons crop out around El Laco volcano in the Central Andes (Fig. 2A; Alva-
Valdivia et al., 2003; Naranjo et al., 2010). Together with the iron ore deposits of
Kiruna (e.g. Jonnsson et al., 2013) they represent worldwide unique examples for very
strong local magnetic anomalies, which may be comparable to that observed in some
areas of the southern Noachian highlands of Mars (Connerney et al., 2005; Lillis et al.,
39  2013).

The selected area with an extension of 0.2 x 0.4 km is situated at the northern
margin of the El Laco Sur outcrop (23°50′17′′S; 67°29′27′′ W; 4720 m elevation; Figs. 1
and 2), at the transition between magnetite-bearing ores and early Pleistocene
andesitic lava flows which are partly covered by pyroclastic deposits with up to 5 m
thickness. The mapping area has not been modified by iron ore mining.



Sernageomin (*Servicio Nacional de Geología y Minería* of Chile; Naranjo et al.,
2010) performed aeromagnetic surveys and constructed maps of the anomaly. They
reflect a dipolar anomaly according to the isodynamic lines of the map with intensities
in the order of ±150 nT (Alva-Valdivia et al., 2003). However, there is no information
available concerning the tracks of these aeromagnetic surveys: spacing and altitudes
above ground, because of which, the data are only considered for completeness.
Besides, on ground magnetic surveys were not done.
Fission-track dating of apatite grown within the magnetites gave an age of 2.1 ±
0.1 Ma (Maksaev et al., 1988). A whole rock age of the andesite from the host rocks of
Pico Laco is 2.0 ± 0.3 Ma (Gardeweg and Ramírez, 1985).
The origin of the magnetite bodies has been strongly debated. A combined
magmatic and hydrothermal origin was proposed by Alva-Valdivia et al. (2003), Sillitoe
and Burrows (2002), and Velasco and Tornos (2012). This is based on the fact that field
and petrographic evidences suggest that some magnetites have a primarily magmatic
texture, whereas others show features which indicate a formation during a
hydrothermal triggered re-emplacement of andesitic lava flows. Trace element
compositions of the latter magnetite type are not compatible with a magmatic origin.
For example, Dare et al. (2014) document that these magnetites are characterized by
high Ni/Cr ratios, depleted in Ti, Al, Cr, Zr, Hf and Sc, and show an oscillating zoning of
Si, Ca, Mg and rare earth elements. In contrast, oxygen isotope data ($\delta^{18}$O of +2 to +4)
of many magnetites support a magmatic rather than hydrothermal origin (Jonnsson et
al., 2013).
Microscopy studies under reflected light as well as temperature dependent
susceptibility measurements and isothermal remanent magnetization (IRM) acquisition
show that low Ti-magnetite and/or maghemite are the magnetic carriers (Alva-Valdivia
et al., 2003). Sometimes ilmenite-hematite minerals appear in significant amounts.
Grain sizes range from a few microns up to several millimetres. Hysteresis
measurements of Alva-Valdivia et al. (2003) of seven ore samples from El Laco Sur
point to pseudo-single-domain status and show a large range of Koenigsberger ratios
(Q-ratios from 0.02 to >1000).
Paleomagnetic data show distinct local declinations indicating a complex
crystallization history, probably during different geomagnetic field orientations (Alva-
Valdivia et al., 2003).
SITE 2 "Pali Aike". Lava sheets with volcanic spatter cones represent a common feature
in many areas of the surface of Mars (Kereszturi and Németh, 2012; Robbins et al.,
2013). On Earth such volcanic rocks often exhibit distinct magnetic anomalies (e.g.
Bolos et al., 2012; Urrutia-Fucugauchi et al., 2012). However, only few examples have
been mapped with high resolution (e.g. Cassidy and Locke, 2010).
Thus, an agglutinated spatter cone of 170 m diameter and surrounding Quaternary
lava sheet of the Pali Aike Volcanic Field (PAVF) in southernmost Patagonia (Figs. 1 and
3; Skewes and Stern, 1979) has been selected as potential martian analogue. The well-
preserved morphology and stratigraphy indicates an age of approximately 1.0 Ma
when considering the succession of various nearby volcanic formations for which ages
of 0.16 to 1.5 Ma have been reported (Mejia et al., 2004). The investigated crater is
partly filled by pyroclastic material, eolian sediments as well as blocks and detritus,



which have been collapsed from the eastern inner crater wall.
The mapping site (52°06'43''S; 69°42'28''W; 227 m elevation) covers an area of
400 x 400 m, including the crater and its surroundings (Fig. 3A).
SITE 3 "Monturaqui". Impact craters represent a very frequent feature on the Mars
surface (Lillis et al., 2013). Depending on e.g. size, target rocks, impactite composition
and possible hydrothermal processes they can be characterised by distinct and
complex magnetic signatures (e.g. Osinski et al., 2013). On Earth, large impact craters
are strongly eroded. In addition, some of them are covered by vegetation or modified
by anthropogenic influences. A Late Pleistocene simple type impact crater in the
Atacama Desert of northern Chile was selected for this case study (Fig. 1). The crater
was discovered in 1962 from aerial photographs and firstly described by Sánchez and
Cassidy (1966). It is located at latitude 23°55'40''S and longitude 68°15'42''W at an
elevation of 2984 m (Ugalde et al., 2007), has a diameter of 370 m and is 34 m deep
(Fig. 4 A, B, C), and was formed during the Quaternary (660 ± 90 kyr BP; Ukstins Peate
et al., 2010). Due to the arid climate, it remained morphologically well preserved. The
crater has remarkable morphological similarities to the Bonneville impact crater on
Mars, which was explored by the Spirit rover of NASA (Grant et al., 2004). Monturaqui
target rocks include Jurassic granites cut by some mafic dykes. Both rock types are
overlain by a several metre thick sheet of Pliocene ignimbrites. Tiny Fe-Ni-Co-P
spherules, all bound in impact glass, have been found within the ejecta blanket. They
suggest an iron meteorite as impactor (Bunch and Cassidy, 1972; Kloberanz, 2010).
SITE 4 "Bahía Glaciares". Plutonic rocks and layered intrusions form significant parts of
the martian crust (e.g. Francis, 2011) and analogues on Earth (McEnroe et al., 2004
and 2009). These rocks may have the capacity to store remanent magnetic signatures
that can be used to distinguish between different magmatic rock types during future
rover-based magnetic surveys. The Patagonian Batholith in the southernmost Andes
provides a good example of continental crust formation on Earth and other planets
(Behrmann and Kilian, 2003; Diaz-Michelena and Kilian, 2015). A small mapping area
of 20 x 6 m (120 m$^2$) was defined on a Cretaceous granite (Fig. 5; Hervé et al., 2007),
which is cross-cut by several mafic North-trending (~ 5 °N) more or less parallel mafic
dykes (52°48'28''S; 73°14'10''W; 11 m a.s.l.). This area was chosen because it is a
good example of very low intensity magnetic contrast in a small extension, where
transition between alternating mafic and felsic outcrops appears at a centimetre-
scale.
**2.4    Additional analyses**
The magnetic field surveys have been complemented with other rock analyses to
improve the interpretation of the magnetic signatures of the surveys. Even though the
detailed analysis is out of the scope of this work, the types of measurements are
briefly described because they support partially some of the conclusions of the work.
Firstly, a macroscopic description of the rock types and mineral components has
been done at each site. Representative rock samples were collected along the tracks
for macroscopic investigation and future analyses in the laboratory. For instance, the
samples from Pali Aike (Site 2) and Bahía Glaciares (Site 4) have been analysed with
polarization and refracted light microscopy of thin sections of the rocks. Texture and



grain sizes of samples from el Laco (Site 1) have been also investigated with a scanning electron microscope (Leo 435 VP, Geology Department, Trier University). The mineral composition of granites and amphibolitic dykes of Bahía Glaciares (Site 4) have been analysed by an X-ray diffractometer (Siemens D500, Geology Department, Trier University).

Hysteresis properties of representative samples from Pali Aike (Site 2) and Bahía Glaciares (Site 4) have been characterized magnetically at room temperature by means of a vibrating sample magnetometer at the Space Magnetism Laboratory of INTA, Spain. Magnetic susceptibilities have been also measured with a MS-2 susceptometer by Bartington along the transects at Site 4.

## 3. Results

The comparative performance and results of the magnetic surveys with MOURA and G-858 magnetometers are described below. In this context some of the individual capabilities of the sensor will be also discussed.

### 3.1 SITE 1 "El Laco"

The surveys were performed with both instruments using continuous and discrete measurement modes. During the surveys the temperature was ranging from 5°C to 27°C. The transects were georeferenced with the GPS. A Matlab code was used to combine, interpolate and merge the magnetic anomalies and tracks (Fig. 2). The vertical magnetic gradient has been measured at one point where magnetite-bearing ores crop out. Fig. 2B shows an exponential increase of the magnetic anomaly from 1.9 m altitude above the ground down to the rock surface (from + 3,000 to +23,000 nT). The gradient field calculated from the vector components of both vector sensors shows a strong negative vertical component ($m_z$ = -11.7 A/m) for this point, which has been modelled by a local shallow superficial dipole with a volume of 0.1 x 0.1 x 0.4 m, an inclination of -62° and a declination of -67°.

The magnetite-bearing outcrops exhibit very high positive magnetic anomalies from 30,000 to >110,000 nT while surrounding andesitic lavas and pyroclastic material have much lower positive anomalies (+100 to +2,000 nT). The magnetic anomalies across outcrop transitions between andesites and magnetite-bearing ores have been measured with MOURA in a discrete mode directly on the ground (Fig. 2C) and with a continuous mode (Fig. 2D). The differences in the andesites anomalies intensity between these two measurements are related to the distance between the sensor and the ground surface (0 and 30 cm). The continuous measurements show a large variability of the magnetic anomalies along the ore-bearing outcrops related to either heterogeneous ore compositions or slight variations of the sensor distance from the surface.

During G-858 surveys the magnetometer became often saturated when high local magnetic anomalies were reached (>80,000 nT). Local surveys with a higher spatial resolution in a continuous mode showed that this situation appeared very frequently and thus did not permit a complete high-resolution survey with this instrument. The surveys with MOURA were not affected by such saturation since this magnetometer

has an extended range mode (auto) that doubles the nominal range to ± 130,000 nT
per axis, and thus, allows measurements up to higher field intensities as it has been
suggested partly for the martian surface. Despite this problem with the G-858 the
scalar magnetic maps of both magnetometers are similar (comparison in Figs. 2E and
2F). Of relevant importance is that MOURA magnetometer allows the identification of
the component that saturates.
Since MOURA magnetometer provides vector magnetic data, it is possible to
determine the orientation of the field in the area. This is shown in the rosette of Fig.
2G together with the paleodeclinations of other rock samples from El Laco Sur
determined by Alva Valdivia et al. (2003).
**3.2. SITE 2 "Pali Aike"**
A dense grid was performed over the depicted surface with G-858 magnetometer (Fig.
3A). In this case, MOURA measurements have been performed with the discrete mode
(Fig. 3B) to obtain well-referenced vector data. During the survey, the temperature
was oscillating between 7 to 15° C.
Fig. 3B compares an interpolated magnetic anomaly map measured with G-858
with discrete points from MOURA that are illustrated with a colour code. Both data
sets match very well. A 3-D view of the interpolated magnetic anomalies mapped with
G-858 is shown in Fig. 3C. It documents the very high positive anomalies of the crater
rim (up to +12,000 nT). A W-E transect of the crater and its surroundings, and its
geological features shows two pronounced positive magnetic anomalies centred at
both sides of the crater rim where the agglutinated spatter have been mainly
deposited as pillow-like blocks of metre size (Fig. 3D).
Vector information obtained with MOURA magnetometer at the individual points
along the crater rim and crater infill is illustrated in Fig. 3E. In general the predominant
declination in all the measurements taken on consolidated lava blocks and the
sedimentary infill is around 355°N (white arrows in Fig. 3E). However, anomalous
deviations have been detected in the Eastern and Southern part of the crater (red
arrows in Fig. 3E), on single basaltic lava blocks, which have removed into the crater
during a post-eruptive collapse of the inner crater wall.
MOURA has two magnetometers at a small distance of 10 mm between them.
They can be used also to measure some of the components of the gradient of the field.
Fig. 3G shows the derivatives respect to z of the components Bx, By and Bz of the field.
In good agreement with the previous conclusion, the gradient seems to have a
homogeneous direction all over the crater with the exception of measurements on
single lava blocks that have been removed and re-orientated during the collapse of the
wall.
For, all these data it has been taken into account the tilt angle of MOURA apart
from the deviation respect to the North taken with the GPS. This has been possible due
to the fact that MOURA has a tilt angle sensor to measure the deviation from the
horizontal. This sensor has been used in this example to derive a gravity contrast along
the transect within the crater and along its rim which is illustrated in Fig. 3F. Highest
values occur along the eastern and western crater rim whereas lowest values are
measured at the western eolian sedimentary crater infill. This relationship is shown in



the W-E transect of Fig. 3D.

### 3    **3.3. SITE 3 "Monturaqui"**

In this example we performed a dense grid of the centre of the crater as well as the
northeastern, eastern and southern part of the crater rim. An interpolated map shows
very slight magnetic anomalies (< 50 nT) within the crater (Fig. 4B) whereas more
pronounced local negative and positive anomalies from -400 to > +600 nT occur within
several meters along the crater rim indicating the existence of metre-sized dipoles. The
anomalies are not related to outcrops of exposed granitoids and ignimbrites (Fig. 4C).
A higher-resolution mapping was performed at a local area of around 10 x 20 m at
the north-eastern crater rim with both magnetometers in a continuous mode with G
858 and by discrete points with MOURA. Both magnetometers show relatively high
positive and negative anomalies ranging from -3,500 up to >+6,000 nT (Fig. 4D). This
local field of anomalies indicate the existence of not exposed but near-surface metre-
sized dipoles. The anomalies measured with both magnetometers are compared in an
X-Y plot of Fig. 4E and indicate a correlation of $R^2$ of 0.81.
The local mapping area at the north-eastern crater rim is characterised by
pronounced local topography changes in the range of ± 5 m of elevation. Fig. 4F
illustrates that more pronounced negative anomalies occur at topographic lows. This
relationship between lower topography points and higher positive anomalies (and vice
versa) was also measured with the B1 and B2 sensors of MOURA and is shown in Fig.
4G. This figure documents also the good correlation between both MOURA sensors, B1
showing on average + 1,500 to +2,000 nT higher values than B2 related to the fact that
B1 is 10 mm nearer to the ground surface.

### 26    **3.4 SITE 4 "Bahía Glaciares"**

A 20 x 20 m area was mapped with G-858 and MOURA magnetometers along seven
high-resolution tracks perpendicular to the dykes (Fig. 5B to 5D) using both continuous
and discrete modes. The spacing between the lines is approximately 80 cm and the
distance between individual measurement points along the lines range from 5 to 10
cm. All the lines show similar patterns, which allow performing an interpolated map of
the area: Figs. 5C and 5D show that the dykes clearly contrast with the granites by
slight positive anomalies.
Fig. 5E shows one of the high-resolution transects where the magnetic signatures
have been measured with both magnetometers and a M2 Bartington device for
susceptibilities. The dykes exhibit very clear but weak positive anomalies with respect
to the granite in the range from +20 to +80 nT. Both magnetometers show similar
patterns. Overall slightly higher values (around 30 nT) of MOURA data are attributed to
the fact that they have been performed directly on the rock surface whereas those of
G-858 magnetometer are measured at a certain distance to the surface (25 to 30 cm).
The susceptibility transect shows a very sharp transition at the interfaces between the
granite and dykes, the latter having around $70 \times 10^{-6}$ SI higher values on average.
The magnetic anomalies and the susceptibility data show laterally displaced (Fig.
6A, B). This reflects the eastward tilt of the dykes which have dipping angles of 50 to
80° and indicates that the uppermost 2-3 m of the mafic dykes are also integrated




within the anomalies measured with both magnetometers while the susceptibility
shows only the information of the first centimetres below the surface.
Detailed petrographic studies including electron microprobe and XRD analysis
show that both granites and amphibolitic dykes do not contain magnetite, but instead
include ferromagnetic monoclinic C4 pyrrhotite as magnetic carrier. Areal microscopic
mapping of pyrrhotite in thin sections of the different dykes and the granite and XRD
analyses of the different rocks indicate a content of 1 to 4 vol.% pyrrhotite with grain
sizes ranging from <5 to 150 µm. There is a good correlation between the pyrrhotite
contents of the different dykes and the amount of the positive anomalies (Fig. 5E).
**4.0    Discussion**
The results of the different ground magnetic surveys of both magnetometers are
discussed with respect to the appropriateness of the different instruments and the
relationship to mineralogical and magnetic properties of the exposed rocks. In
particular the potential of high-resolution detection of weak magnetic contrast
between different surface rock types is considered.
**4.1 Instrument Performance**
Final processed data from both instruments show a very good correlation in intensity
of magnetic anomalies for the overall measurement range between -2,000 nT and
100,000 nT (Figs. 2E, 2F, 3C, 4E and 5E).
The stability of both instruments has been appropriate for the different
surveys. G-858 MagMapper shows a better thermal stability, which can be observed
during faster temperature variations during the dawn and dusk, when the transducer
may experiment thermal variations up to 0.1 °C/min. The simultaneous measurement
of the temperature and the magnetic field diminishes this problem, which can be
neglected in areas with magnetic anomalies > 100 nT, but the error can be significant
(1 %) in low contrast anomalies (1 nT), also due to the resolution of MOURA
instrument. Other ways to compensate the temperature effects could improve these
errors (Díaz-Michelena et al., 2015).
Regarding the dynamic range, both magnetometers have also casted
appropriate results in most of the cases. The limitation in this feature affects in a
different way the response of both instruments. G-858 is affected in the measured
modulus of the field, while MOURA is affected separately in every axis. This is an
advantage since it could provide useful data in two directions despite of saturation in
the other axis. For example, at Site 1, the huge intensity of the anomalies makes it
impossible to map them with G-858, while MOURA can measure them in the auto
mode, when the maximum offset is applied.
At the El Laco site MOURA surveys were performed with discrete and
continuous modes. The continuous mode enabled a higher resolution and an easier
performance but it may include a shifting by slight variations of the distance between
sensor and the ground. The extreme high gradient at this site causes a pronounced
fluctuation of around >5,000 nT when altitude of the sensor changes from 25 to 30 cm.
This can be avoided by discrete measurements directly on the ground, which also



enable a better orientation control of the vector sensor. In the case of highly positive anomalies and in combination with high Q-ratios (remanent versus induced magnetic signatures) of the surface rocks, the vector measurements may also have the capability for paleomagnetic implications which is extremely important for planetary exploration.

Surface rock alteration processes which modify the magnetic signatures have influence on limited areas. In particular, the related mineral transformations processes are a direct consequence of the contact of the rocks with the hydrosphere and atmosphere, and their influence depth is limited to several tens of metres. This fact together with the exhumation processes offers often the possibility to correlate the measured direction of the magnetization with the coetaneous paleomagnetic field. In the case of Mars, where the main source of field is the remanent magnetization, oriented measurements does not only contain information on the carriers and the possible alteration effects suffered by them, but also record the paleomagnetic field orientation. This possible tool has been applied to remanence dominated sites which is further discussed in section 4.2.

The sensor head orientation is also a matter of discussion. Despite the good signal-to-noise ratio presented by the scalar magnetometer, it is affected by the relative orientation of the head and the magnetic field vector. This is highly improved in a 3-axes magnetometer like MOURA.

Another consideration is the gradient immunity and capability to derive a gradient of the field of the instruments. On the one hand, MOURA instrument presents a better gradient immunity, which makes it very suitable to map areas with high frequency patching of the signatures. For example, it is very appropriate to perform decimetre-scale resolution mappings like in the cases of El Laco, Monturaqui and Bahía Glaciares. G-858 presents troubles with not so high gradients (>20,000 nT).

On the other hand, the inclusion of a second head (and therefore to have two 3-axes magnetometers) in MOURA instrument offers the capability to better understand the characteristics and depths of the sources. This cannot be applied to deep and extended magnetic sources, because the distance between the two magnetometers is very small (10 mm) but it is useful to analyse near surface heterogeneities and will be discussed in section 4.2.

### 4.2 Capacity for high resolution mapping with tracing of mineralogical and geological characteristics

High-resolution ground surveys may indicate compositional variations in soils and/or uppermost crustal rocks, depending on the magnetic contrast between different exposed rocks and the intensity of active magnetic field (Gobashy et al., 2008; Hinze et al., 2013). The exploration capacity of the commercial G-858 and MOURA magnetometer for extra-terrestrial high-resolution mapping is discussed in the following for the different investigated sites.

**El Laco**

At this site the intensities in the magnetic anomalies range from 0 to +110,000 nT (Figs. 2C to 2F) which is unique compared to other magnetic mapping results on Earth (e.g. Hinze et al., 2013). The magnetic contrast at the surface transition between andesitic rocks and magnetite-bearing ores is very sharp and extremely high. A change from



+1,200 to +80,000 nT appears in less than a metre distance. In general, MOURA data
show a better definition of the surface rock transitions and local variabilities in the
areas with outcrops of magnetite-bearing ores compared to G-858. Furthermore, G-
858 could not register some of the anomalies because its response was occasionally
saturated (110,000 nT) (Fig. 2D). Some variations in areas with exposed ores (in the
order of ± 5,000 nT) could have been caused by slight changes in the sensor distance
from the ground during a continuous measuring mode (see Chapter 3.1), the major
variations are probably related to the heterogeneous composition and locally distinct
magnetic behaviour of the magnetite-bearing ores. The texture of magnetites in the
ores indicates in part a primarily volcanic origin, but also frequent recrystallization
during later hydrothermal processes can be observed. It is likely that the hydrothermal
crystallization took place over a longer period of the Early Quaternary which may have
also included magnetic reversals (e.g. Alva-Valdivia et al., 2003; Naranjo et al., 2010).
Our field observations and laboratory analyses of collected samples indicate a large
scatter in the grain sizes, the porosity content as well as the relative amount of
additional apatite (non-magnetic) and pyroxene in these rocks. These features might
explain the observed variations. Other variables reported by Alva-Valdivia et al. (2003)
include hysteresis parameters and highly variable Q ratios (from 0.01 to >5,000)
indicating a wide range of individual properties of the magnetic carriers, compatible
with pseudo-single-domain up to multi-domain status. Therefore, we attribute the
observed huge variability in magnetic anomalies to local variabilities in the behaviour
and magnetic properties of magnetites in near-surface rocks.
In areas where andesitic lavas are exposed, field surveys show only low
fluctuations of the positive anomalies (Fig. 2C and 2D). This let us to hypothesize that
there are no underlying local ore bodies and the lava flows have relatively
homogenous compositions.
Measurements with the MOURA vector magnetometer show a clear northward
declination between 350° to 10° N (Fig. 2G). This value is similar to the present
declination of the IRGF at this site (3° N) and can be explained by a very strong induced
magnetization consistent with the very high susceptibilities, but relatively low Q ratios
(0.01) measured in six of seven samples from El Laco Sur by Alva-Valdivia et al. (2003).
**Pali Aike**
Magnetic surveys have been performed at some volcanoes world-wide with different
spatial resolution, e.g. from Australia (Blaikie et al., 2012), New Zealand (Cassidy and
Locke, 2010) and Italy (Okuma et al., 2009). In general these case studies show more
and less positive magnetic anomalies (up to a few thousand nT) depending on the
composition of the volcanic rocks, its cooling history and the single versus multi
domain status of their magnetites (Clark, 1997). Our example of a small crater (170 m
diameter) and its surroundings at the Pali Aike Volcanic Field was performed with both
magnetometers and with a higher spatial resolution than the previous studies. The
transect across the Pali Aike crater shown in Fig. 3C and 3D has a spatial resolution of
30-50 cm in-between individual measuring points and thus provides a very good
differentiation of different kinds of exposed rocks within the uppermost 1-2 m.
Despite the relatively high intensity of the IGRF at Pali Aike (+ 31,000 nT; Fig. 1) the
transect across the crater is characterised by very high positive magnetic anomalies of



up to + 12,000 nT. These anomalies are mostly pronounced along the crater rim where
metre-sized melt spatters have been deposited and cooled down (Fig. 3D). The very
strong magnetic signature can be explained by the fact that these lavas contain very
frequent tiny magnetite crystals with single domain characteristics in their glassy
matrix. The positive anomalies become much lower towards the crater infill and the
outer slopes of the crater. An increasing amount of pyroclastic deposits with
reorientations during the deposition processes on the steeper slopes of the crater
could have reduced the integrated magnetic anomaly of these components. The
relatively low local anomalies measured within the crater can be also explained by
such multiple re-orientations of magnetic carriers during local redistribution processes
including fluvial and eolian activities.

12       Fig. 3E shows arrows for the declination calculated from the vector data of
MOURA. The values of all measurements on consolidated lava blocks (white arrows in
Fig. 3E) range from 352° to 360° N. These orientations may reflect either the induced
present magnetic field or paleofield directions, or a combination of both, depending
on their remanence and related Q ratios which reflect often the single versus
multidomain status of the basalts. Hysteresis measurements of 25 basaltic lava
samples from our mapping area indicate relatively high Q-ratios of 50 to >500
suggesting a strong remanence. Comparable basaltic rocks from other sites worldwide
are also characterized by a very strong remanence and a predominant single domain
status (Day, 1997; Dos Santos et al., 2015; Dunlop, 2002; Zhao et al., 2006). Our
measurements which have been performed directly on detached lava and scoria
blocks, clearly collapsed from the inner crater wall, show multiple orientations. They
are indicated by red arrows in Fig. 3E and include easterly and westerly declinations.

The present field which has a declination of 12°N at Pali Aike, the MOURA field
vectors as well as several paleodeclinations from different old Pleistocene lavas of Pali
Aike (including magnetic reversals; Mejía et al., 2004) have been compiled in Fig. 3H.
The estimated age of the investigated cone is approx. 1.0 Ma which suggests a normal
global field for that time. MOURA declinations, ranging from 352° to 360° N, are in a
very good agreement with such a normal declination as well as other normal
declination constrained for other lavas from the Bruhns magnetic period (Mejía et al.,
2004) rather than the present field declinations which contrast by +15°. This result
indicates that MOURA magnetometer could provide paleodeclinations when rocks
have a very high remanence (high Q ratios).

**Monturaqui impact crater**
Planetary impact craters can be characterised by a variety of magnetic anomalies
which are related in particular to distinct magnetic carriers of the target rocks (e.g.
mafic versus felsic or sedimentary) and the sedimentary crater infill as well as the
compositions of the impactor, impact-induced melt/glass and/or impact-related
hydrothermal mineralization and/or demagnetization (e.g. L'Hereux et al., 2008;
Langlais and Thébault, 2011; Osinski and Pierazzo, 2013; Pilkington and Grieve, 1992;
Prezzi et al., 2012).

44       At the relatively small Monturaqui crater, a coarse grid of magnetic mapping with
spacings of approximately 70 m in-between single measuring points have been



previously performed with a caesium magnetometer in the crater and its surroundings
by Ugalde et al. (2007). The published interpolated map shows only very low magnetic
anomalies in the range of less than ± 200 nT with slightly higher values in the southern
and eastern sector of the crater rim. Our grid was measured with a continuous mode
of the G-858 and provides a much higher resolution for the central part of the crater as
well as its northern, eastern and southern rim and slopes (Fig. 4B). Our data show
similar low magnetic anomalies of ± 50 nT for the crater floor as measured by Ugalde
et al. (2007), whereas the eastern and north-eastern crater rim is characterised by
much stronger local anomalies from -500 to >+ 600 nT. The transitions of the outcrops
of granitic and ignimbritic target rocks shown Fig. 4C cannot be depicted by the distinct
anomalies, while two northwest to southeast-trending dykes seems to responsible for
some anomalies at the eastern crater rim (Fig. 4B). In general, the crater rim is
characterised by a patchwork of pronounced local positive and negative anomalies
which can be caused by small-scale near-surface dipoles which are probably not
exposed. Not exposed fragments of the impactor, for which a diameter of ~15 m has
been modelled (Echaurren et al., 2005), represent a potential source for these
magnetic anomalies. Fe-Ni spherules which have been found in impact melt/glass
fragments along the eastern and western crater rim has been classified as a Group I
Coarse Octahedrite (Bunch and Cassidy, 1972; Buchwald, 1975) and contain
schreibersite, cohenite, fossil taenite, and kamacite as major components as well as
some troilite. Laboratory analyses of these components show both high remanent
magnetization as well as very high susceptibility (Ugalde et al., 2007; Ukstins Peate et
al., 2010).
The very pronounced negative to positive anomalies from -2,000 to +5,000 nT
which have been mapped in a local area of 50 to 100 m extension at the northeastern
crater rim (Fig. 4D) also require near-surface rocks with strong dipoles. Not exposed
metre-sized fragments of the iron-bearing impactor represent the most likely
explanation. In this local area topographic highs are formed by ignimbrite blocks of
low-magnetic signature which has been ejected from the crater and deposited on the
rim during the impact event. Topographic highs formed by these blocks cause a higher
distance of the magnetic sensor with respect to the probably underlying fragments of
the impactor while measurements in topographic lows show much higher positive
anomalies (Fig. 4F and 4G).

**Bahía Glaciares**

Mafic dykes within felsic to intermediate crustal rocks often produce pronounced local
positive magnetic anomalies since they include frequent tiny magnetites (e.g. Hinze et
al., 2013). However, in our case study the petrographical investigations indicate that
the investigated dykes have not preserved their original magmatic mineral textures
and the mineral assemblage point to an emplacement and later equilibration of the
dykes under upper greenschist to amphibolite facies conditions (Bucher and Grapes,
2011; Philpotts and Ague, 2009). Granites and dykes do not contain magnetite, but
both contain monoclinic pyrrhotite as magnetic carrier (Dekkers, 1988, 1989; Clark,
1984). This mineral appears disseminated and along veins and has been formed during
hydrothermal mineralization together with Cu and Au enrichments during the
exhumation (Díaz-Michelena and Kilian 2015; Nelson, 1996; Schalamuk et al., 1997).





Despite the lower potential of pyrrhotite to produced magnetic anomalies both magnetometers (MOURA and G-858) clearly show high resolution and slightly positive magnetic anomalies (+30 to +80 nT) as well as higher susceptibilities of the dykes compared to the granites (Fig. 5E). The surface transitions between dykes and granites are characterised by very sharp anomalies within a decimetre scale. The amount of pyrrothite (1 to 4 vol. %) which has been quantified in samples of the granite and seven dykes shows a very good correlation with the intensity of the positive anomaly (from +160 to +220 nT; Fig. 5F). This result confirms the potential of both magnetometers to explore local mineral enrichments which are often produced by hydrothermal processes associated with gold and copper enrichments (Direen et al., 2008).

Hinze et al. (2013) show examples of mafic dykes in felsic rocks where different dyke geometries cause distinct shapes of local magnetic anomalies across dykes. For the investigated Bahía Glaciares site Fig. 6 illustrates asymmetric behaviour of the magnetic anomalies along lines which have been measured across dykes which dip with between 50 to 80°. All anomalies are slightly displaced towards the shallower dipping site of the dykes indicating an integration of the magnetic signature of the uppermost 2-3 m of not exposed dykes. This fact together with the difference in contrast obtained with the magnetometers and the susceptometer (Fig. 5) points out that the magnetic field measurements average large volume sources and can lead to wrong conclusions if used as a quantitative mineralogical marker. These results indicate that a multihead (magnetometer and susceptometer) instruments would provide much better results for this purpose.

## 5. Conclusions

Several sites with a huge variability in magnetic anomalies have been analysed. As a first conclusion it can be said that the surface measurement of the sourced field often gives direct information on the composition, petrogenesis and alteration processes of the surface rocks.

For the study, two different magnetometers have been used. On the one hand MOURA vector magnetometer and gradiometer (< 200 g: 72 g instrument + batteries and control PC), developed for Mars exploration, as the demonstration of the technology, and on the other hand a commercial caesium G-858 magnetometer (8 - 9 kg), also used as a reference. The studied areas are considered Mars analogues and they are representative of the intensity range of the expected anomalies on the Red Planet crust.

According to the comparison with the reference instrument, it has been demonstrated that MOURA magnetometer is suitable for the measurement of this range of sourced fields: from <15,000 to >120,000 nT and to reproduce the magnetic contrast of the terrains. Furthermore, MOURA offer vector data of the field and components of the gradient with a significantly lower mass.

The particular conclusions for the four case studies are the following:

1) El Laco magnetite-bearing ore deposits in the Northern Andes of Chile represents a world-wide unique example with extremely high on ground anomalies ranging from



+30,000 to +110,000 nT, which may be comparable to highly magnetic rocks of the
Noachian martian crust.

3        In this case MOURA enabled better results than G-858 due its larger range of
operation (130,000 nT /axis). The declinations measured also by MOURA vectors
represent the active global field due to induced-dominated magnetic rock
properties with very low Q-ratios.
2) A crater in the Pali Aike Volcanic Field, in southern Chile, shows very high positive
8        magnetic anomalies (up to 12,000 nT) of its crater rim caused by frequent tiny and
9        single domain magnetite crystals in the matrix of basaltic lava spatters.

10        Since these rocks, like that of many other comparable volcanic rocks on Earth
and other planets, have high Koenigsberger ratios (Q) and thus are dominated by
their remanence, MOURA vector data have been used to determine the
paleomagnetic orientation during the crater formation around 1 Ma before present.
In addition, the different later reorientations of single lava blocks during their
collapse from the steep inner crater wall could have been constrained by the vector
data.
3) The small Monturaqui impact crater in the Atacama Desert of Northern Chile
18        represents an analogue for many other simple type craters, like Bonneville crater on
19        Mars. The granitoid and rhyolitic target rocks have few magnetic carriers and only
20        week magnetic anomalies. Pronounced anomalies along the crater rim indicate
metre-sized unexposed remnants of the iron-bearing impactor (octahedrite).

Local mapping with a decimetre resolution, with intensities ranging from -2000
to +6000 nT, corroborates the existence of such localised, strong and relatively
small size (1 metre) dipoles (iron meteorite fragments) near the surface.

4) A site within the Patagonian Batholith of the southernmost Andes provides a
window into deeper planetary crustal magnetic signatures. The exposed rocks
include granites and mafic dykes that have been partly equilibrated at lower
amphibolite facies conditions, where all primary magmatic magnetites have been
transformed to iron-bearing silicates, and a later hydrothermal mineralization
produced pyrrhotite as only magnetic carrier.

The freshly exposed transitions between these granites and mafic dykes have
been mapped on a decimetre-scale. Despite the very low magnetic contrast from 20
to 80 nT both rock types could have been clearly distinguished. In addition, the
amount of pyrrhotite, ranging from 1 to 4 vol. %, is well correlated with the positive
magnetic anomalies of the dykes. This documents the potential for mapping of
hydrothermal mineralization processes as well as associated gold and copper
enrichments, even if the magnetic contrast is very low.

**Acknowledgements**:
Authors acknowledge all MOURA MetNet team; in particular, the payloads electronics
engineering laboratory for their work with the magnetometer and V. Apéstigue for the
technical support. This work was supported by the Spanish National Space Programme



(DGI-MEC) through the project AYA2011-29967-C05-01 and the Spanish National Space
Program of R&D Externalization through the project PRI-PIBUS-2011-1105.

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





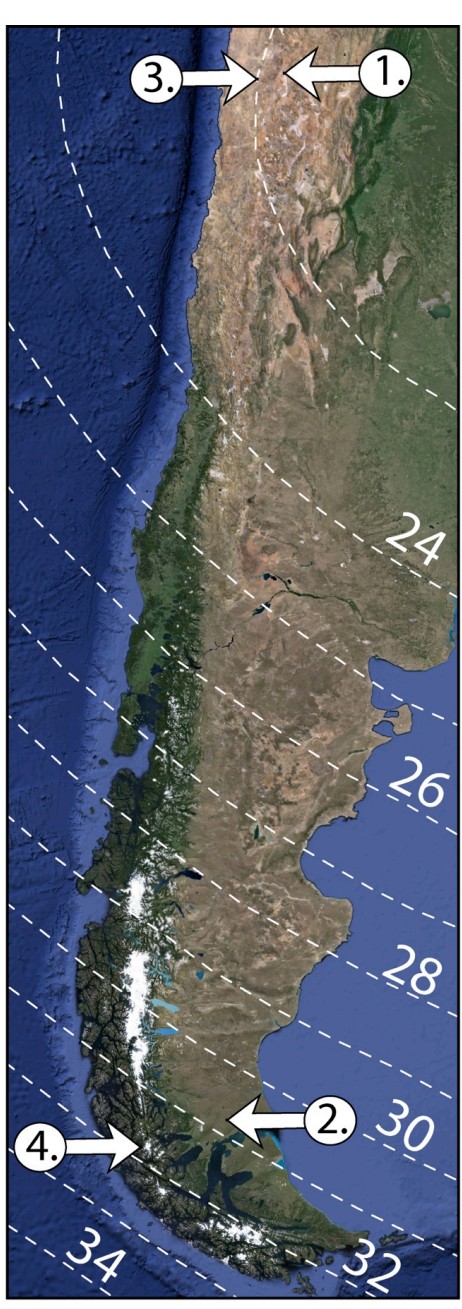

**Fig. 1: Selected sites:** 1: El Laco, 2: Pali Aike, 3: Monturaqui and 4: Bahía Glaciares located in South
America between latitudes 20° and 52°S. The white stippled lines indicates isolines for intensities (in nT
x $10^3$) of the International Geomagnetic Reference field ranging from 23,000 to 32,000 nT at the
different sites.



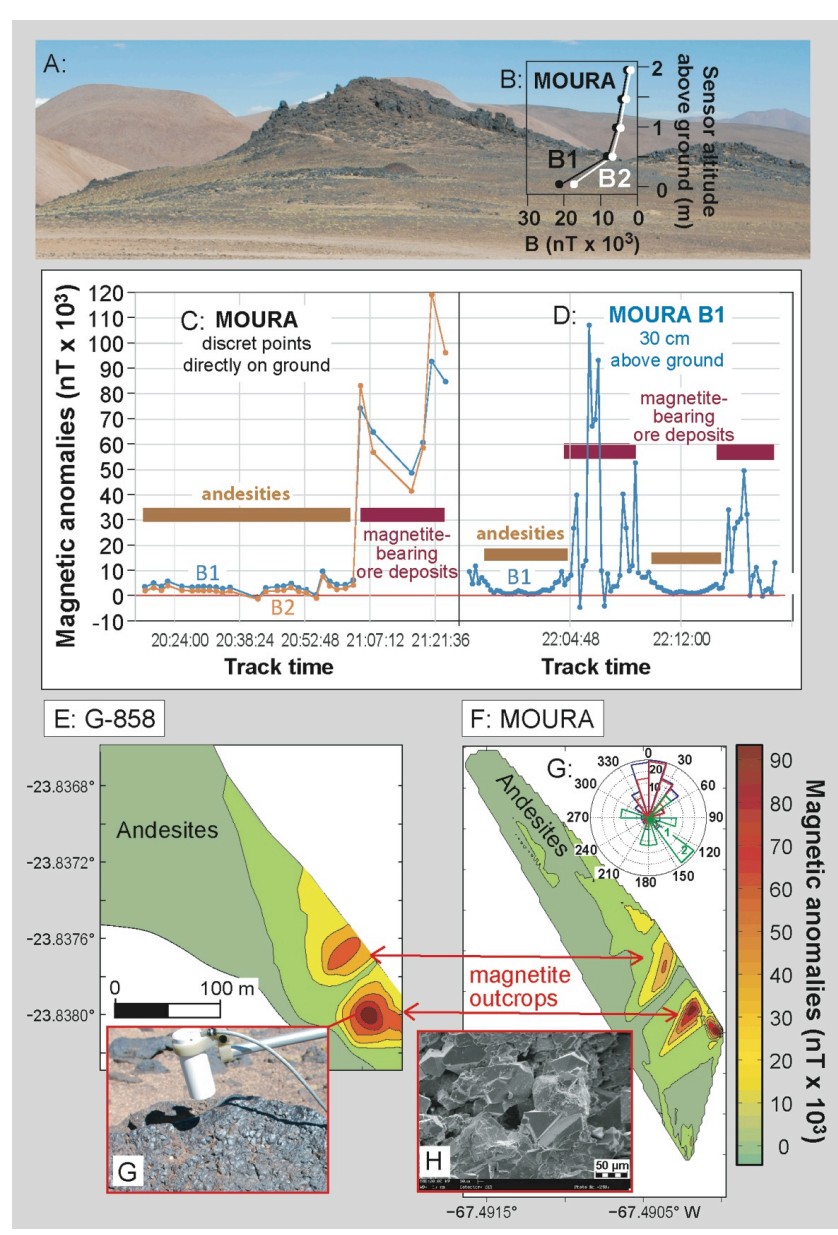

Fig. 2. **Magnetic and geological features of El Laco Sur**. A: A 150 m wide outcrop of magnetite-bearing ores. B: MOURA vector measurements at different distance from the ground. C: MOURA track with discrete measuring points across the transition between andesite and magnetite-bearing ore outcrops. D: MOURA track with the sensor B1 in a continuous mode across different patches of andesites and outcrops of magnetite-bearing ores. E: Interpolated map calculated from survey tracks with G-858 magnetometer. F: Interpolated map based on MOURA mapping. G: Comparison of declination frequencies calculated from MOURA vector sensors (B1 in blue and B2 in red) along a 950 m long track line including 125 data points. Green orientations indicate paleodeclinations determined by Alva-Valdivia et al. (2003). G: G-858 sensor head on a magnetite-bearing rock surface. H: Secondary electron image of the magnetite bearing rock surface.


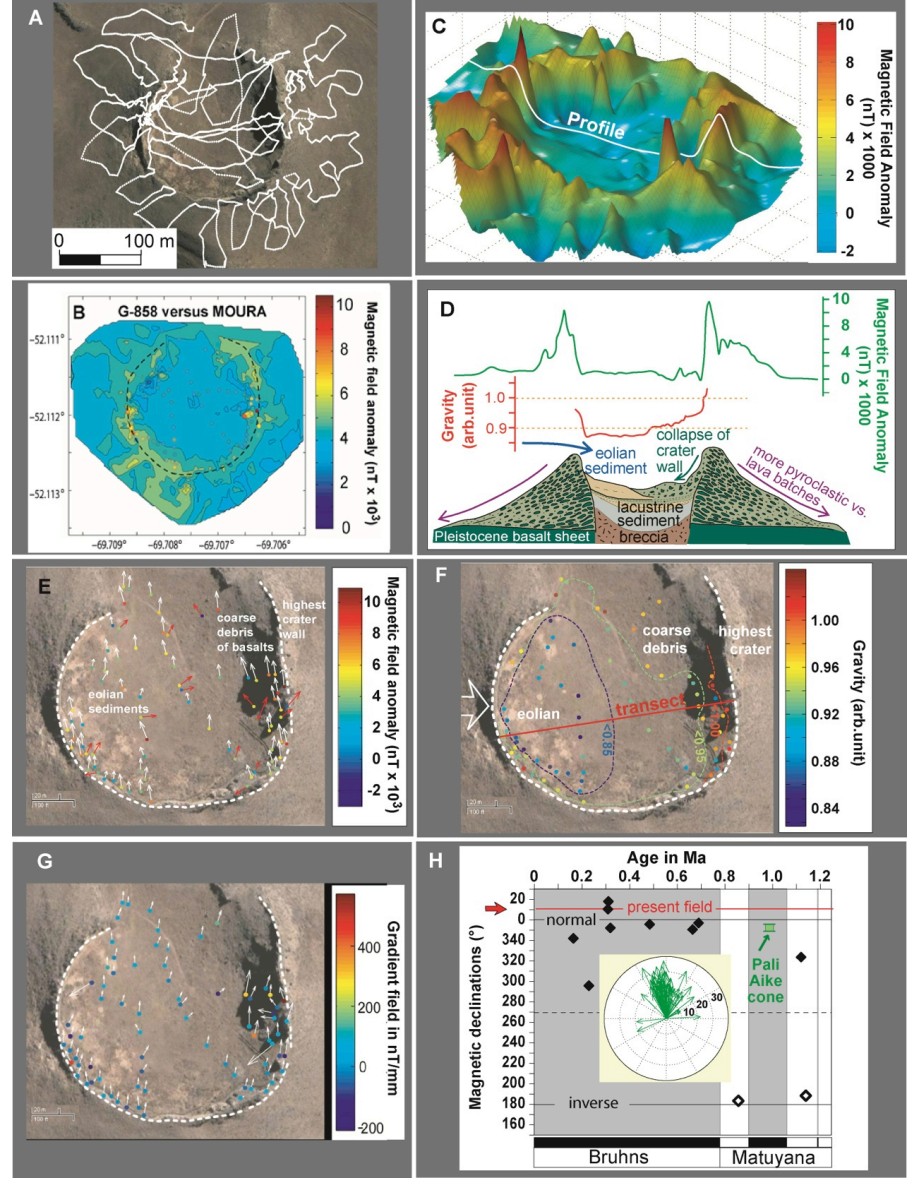

**Fig. 3. Magnetic and geological aspects of a Pali Aike crater and its surroundings.** A: Survey grid with G-858. B: Comparison of the interpolated magnetic anomaly map of G-858 and discrete measurements of MOURA shown in a comparable colour code. C: 3-D view of interpolated magnetic anomalies with the position of the transect shown in D. D: E-W transect across the crater and its geological structure. Magnetic field anomalies are indicated in green and gravity values in red. E: Magnetic anomalies of MOURA and its declination. Red arrows indicate declinations of removed blocks and white arrows that of consolidated rocks. F: $B_{zx}$, $B_{zy}$ and $B_{zz}$ components of the gradient field in nT/mm. G: Gravity measurement points in the crater. A white arrow indicates the direction of typical westerly wind and related eolian deposits in the crater. H: Quaternary paleomagnetic anomalies. Single diamond data points indicate ages and paleoorientations of different old volcanic rocks of Pali Aike determined by Mejia et al. (2004) and a rosette indicates declinations measured with MOURA.





**Fig. 4: Magnetic and geological features of Monturaqui impact crater.** A: Photo of the crater looking towards the North. B: Track (red stippled line) and interpolated mapping grid of G-858. C: Topography and exposed geological units. D: Detailed mapping area at the northeastern carter rim with discrete magnetic data points of MOURA on an interpolated magnetic map of G-858. D: X-Y comparison of MOURA sensor B1 with G-858 data. F: Topography of the local mapping area (shown in D) and areas with stronger positive anomalies at topographic lows marked in red. G: Comparison of MOURA B1 and B2 data and their local elevation.

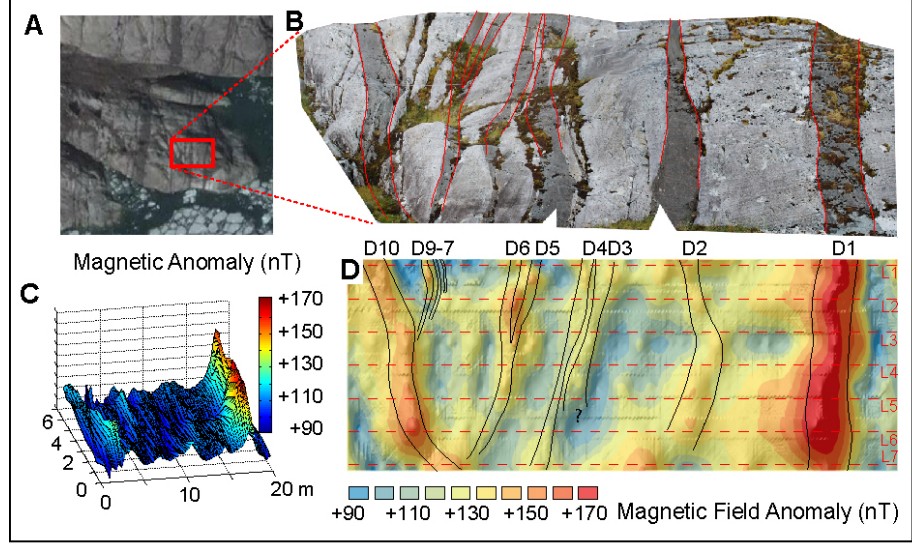

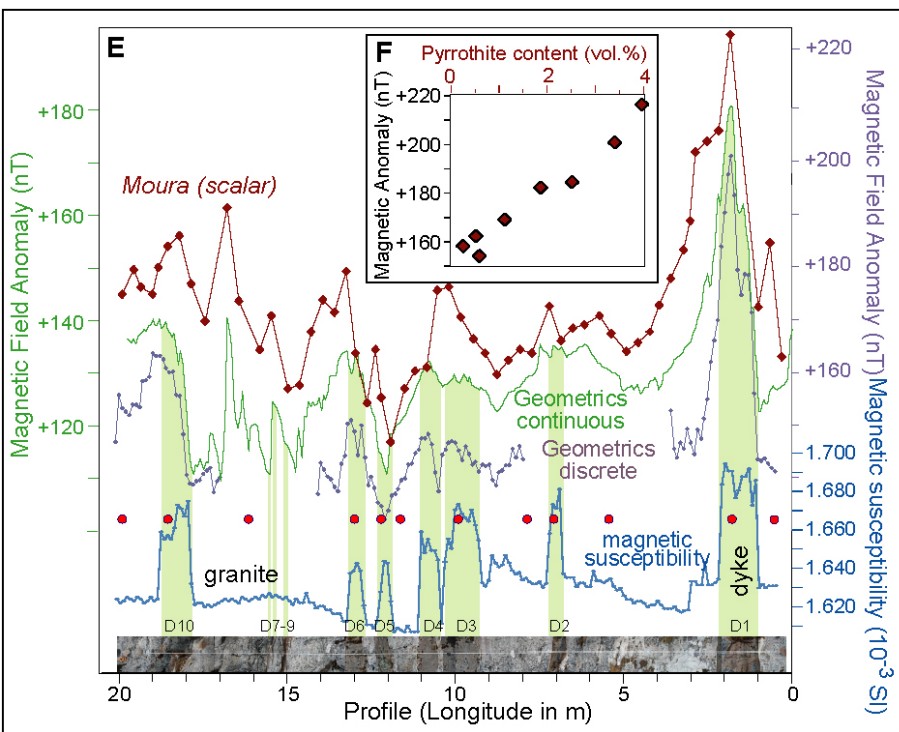

**Fig. 5: Bahía Glaciares mapping area with mafic dykes in a granite:** A: Google Earth view of the
Patagonian Batholith with the local mapping area marked in red. B: Local mapping area with dark dykes
in a granite. C: Interpolated 3-D map of G-858 mapping. D: Magnetic anomalies of the mapping area
with measuring lines L1 to L7. E: Comparison of the intensity of the magnetic anomalies measured by
both magnetometers (Geometrics 858 in discrete and continuous mode, and MOURA) and
susceptibilities, all of them measured in a transect perpendicular to the dykes D1 to D10 (marked in
green). Red dots indicate sample locations along the transect.

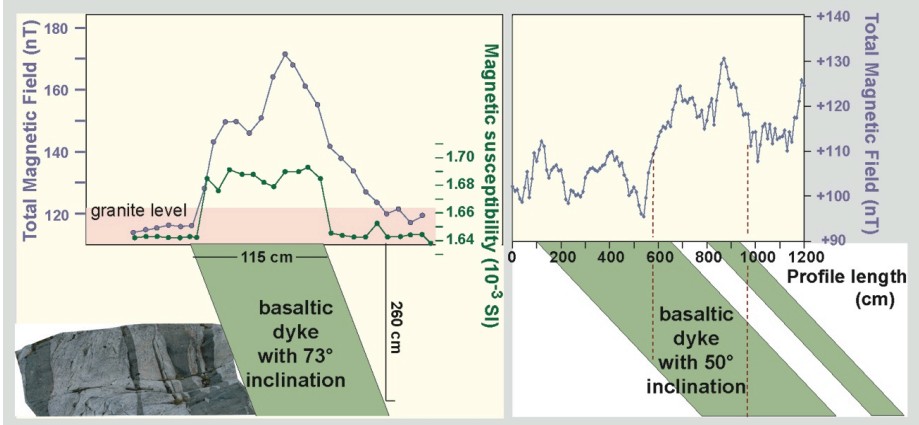

**Fig. 6: Magnetic effects of inclined dykes in perpendicular transects**. A: Asymmetry of the D1 dykes
which is caused by its 70° inclination capturing a local signal up to 260 cm depth. B: A 500 cm wide dyke
with an inclination of up to 50° integrating deeper signatures up to more than 500 m depth.



| Sensor | Scalar magnetometer Geometrics 858 | Vector MOURA magnetometer |
|---|---|---|
|  | Scalar sensor and absolute | Two 3-axis magnetic sensor |
| Dynamic range | 20,000 – 100,000 nT | ±65,500 nT (±130000 nT, autorange mode) |
| Resolution | 30 pT | 0.2 nT |
| Cycle rate | 0.1 s-1 h | 0.1 s |
| Gradient tolerance | >20,000 nT/m | ~1,000,000 nT/m |
| Temperature Drift | <0.05 nT / ºC | $(3.7 - 6.5)\cdot 10^{-6}$ % in gain $(2.0 - 7.9)\cdot 10^{-6}$ % in offset depending on the axis [Diaz-Michelena et al., 2015] |
| Working mode | Continuous and discrete mode | Continuous and discrete mode |

**Table 1.** Summary comparison between MOURA and Geometrics 858 magnetometers.



17          END OF MANUSCRIPT

