# Peer review of "Mars MOURA magnetometer demonstration for high resolution mapping on"

_Geoscientific Instrumentation, Methods and Data Systems, 2015_

## Referee Comment (RC1) · S. Cherkasov (Referee) · 7 Mar 2016

Field measurements clearly demonstrate a suitability of the MOURA magnetometer for magnetic measurements. Nevertheless, when on Mars, MOURA will be not moving, but positioned at some place. So, the expected results will include: a) absolute measurements of the magnetic field (vector) at the point where the magnetometer is located; b) temporal variations of the magnetic field at the point. From the paper, it is clear that the magnetometer will provide data on the absolute magnetic vector with suitable precision. What is unclear, is the precision of the temporal variations' measurements. The data on such variations will be of extreme value for the further analysis. I would advise the author: a) to include in the article brief information on the MOURA's

tests in the artificial magnetic field. Such tests, undoubtedly, have been executed by the author earlier, and they will demonstrate the MOURA's suitability for the magnetic variations' measurements; and b) in the conclusions I would recommend to underline the importance of both, the absolute level, and the variations, measurements, so a reader can better understand the goals of the research.

---

## Referee Comment (RC2) · F. Martín-Hernández (Referee) · 7 Mar 2016

Results shown here are excellent in quality and none can doubt the MOURA magnetometer performance will be a great advance in planetary exploration.

However, when showing the results in the analogues on Earth, the paper lacks order and results are not consistent from one site to the next. In this sense, I will appreciate if for all the analyzed sites have the same protocol as Moturaqui site. A comparison between MOURA and the commercial instrument, a result of both measurements and a correlation between measurements on selected sites.

Modelling the magnetic signal due to subvertical and incline dykes seems out of the

scope of this paper and, although it is interesting as such, increases the chaos showing non-tecnical aspects of the work.

---

## Author Comment (AC1) · 18 Mar 2016

Dear Prof. Martín,

We have read through your comments and have modified the manuscript accordingly. (Text attached with changes highlighted: changes in yellow concern your comments)

We hope that the modifications accomplish with your expectation.

Sincerely, Marina

Mars MOURA magnetometer demonstration for high resolution mapping on terrestrial analogues

[Figure]

Marina Diaz-Michelena1, Rolf Kilian2,3, Ruy Sanz1, Francisco Rios2, Oscar Baeza2

[revised manuscript text omitted]
 all cases with identical single point measurements the correlation between the two instruments and the two sensors of

MOURA has been analysed. The number of differences among the distinct sites has made it possible to demonstrate the versatility of MOURA. Thus, in each study case some of the individual capabilities of the instrument will be highlighted and discussed.

3.1 SITE 1 "El Laco" The surveys were performed with both instruments using continuous and discrete measurement modes. During the surveys the temperature was ranging from 5°C to 27°C. The transects and individual measuring points were geo-referenced with the GPS. A Matlab code was used to combine, interpolate and merge the magnetic anomalies and tracks (Fig. 2). The vertical magnetic gradient has been measured at one point where magnetite-bearing ores crop out. Fig. 2B shows an exponential increase of the magnetic anomaly from 1.9 m altitude above the ground down to the rock surface (from + 3,000 to +23,000 nT). The gradient field calculated from the vector components of both vector sensors shows a strong negative vertical component ($m_z$ = -11.7 A/m) for this point, which has been modelled by a local shallow superficial dipole with a volume of 0.1 x 0.1 x 0.4 m, an inclination of -62° and a declination of -67°. The magnetite-bearing outcrops exhibit very high positive magnetic anomalies from 30,000 to >110,000 nT while surrounding andesitic lavas and pyroclastic material have much lower positive anomalies (+ 100 to + 2,000 nT). The magnetic anomalies across outcrop transitions between andesites and magnetite-bearing ores have been measured with MOURA in a discrete mode directly on the ground (Fig. 2C) and with a continuous mode (Fig. 2D). The differences in the intensity of andesite anomalies between these two measurements are related to the different and slightly variable distance between the hand-held sensor (25 to 30 cm) and the ground surface of single points (0 
[revised manuscript text omitted]
., 2015 b). For static measurements the simultaneous temperature measurement corrects very well the magnetic field data (Díaz-Michelena et al., 2015 a). Regarding the dynamic range, both magnetometers have also casted appropriate results in most of the cases. The limitation in this feature affects in a different way the response of both instruments. G-858 is influenced in the measured modulus of the field, while MOURA is affected separately in every axis. Apart from the extension of the range in modulus, this is an advantage since it could provide useful data in two

directions despite of saturation in the other axis. For example, at Site 1, the huge intensity of the anomalies makes it impossible to map them with G-858, while MOURA can measure them in the auto mode, when the maximum offset is applied. At the El Laco site MOURA surveys were performed with discrete and continuous modes. The continuous mode enabled a higher resolution and an easier performance but it may include a shifting by slight variations of the distance between sensor and the ground. The extreme high gradient at this site causes a pronounced fluctuation of around > 5,000 nT when altitude of the sensor changes from 25 to 30 cm. This can be avoided by discrete measurements directly on the ground, which also enable a better orientation control of the vector sensor. In the case of highly positive anomalies and in combination with high Q-ratios (remanent versus induced magnetic signatures) of the surface rocks, the vector measurements may also have the capability for paleomagnetic implications which is extremely important for planetary exploration. Surface rock alteration processes which modify the magnetic signatures have influence on limited areas. In particular, the related mineral transformations processes are a direct consequence of the contact of the rocks with the hydrosphere and atmosphere, and their influence depth is limited to several tens of metres. This fact together with the exhumation processes offers often the possibility to correlate the measured direction of the magnetization with the coetaneous paleomagnetic field. In the case of Mars, where the main source of field is the remanent magnetization, oriented measurements does not only contain information on the carriers and the possible alteration effects suffered by them, but also record the paleomagnetic field orientation. This possible tool has been applied to remanence dominated sites which is further discussed in section 4.2. The sensor head orientation is also a matter of discussion. Despite the good signal-to-noise ratio presented by the scalar magnetometer, it is affected by the relative orientation of the head and the magnetic field vector. This is highly improved in a 3-axes magnetometer like MOURA. Another consideration is the gradient immunity and capability to derive a gradient of the field of the instruments. On the one hand, MOURA instrument presents a better gradient immunity, which makes it very suitable to map areas with high frequency patching

of the signatures. For example, it is very appropriate to perform decimetre-scale resolution mappings like in the cases of El Laco, Monturaqui and Bahía Glaciares. G-858 presents troubles with not so high gradients (> 20,000 nT). On the other hand, the inclusion of a second head (and therefore to have two 3-axes magnetometers) in MOURA instrument offers the capability to better understand the characteristics and depths of the sources. This cannot be applied to deep and extended magnetic sources, because the distance between the two magnetometers is very small (10 mm) but it is useful to analyse near surface heterogeneities and will be discussed in section 4.2.

4.2 Capacity for high resolution mapping with tracing of mineralogical and geological characteristics High-resolution ground surveys may indicate compositional variations in soils and/or uppermost crustal rocks, depending on the magnetic contrast between different exposed rocks and the intensity of active magnetic field (Gobashy et al., 2008; Hinze et al., 2013). Despite the fact that it is not the primary goal of MOURA instrument, which is part of the instruments suite of a lander, due to its potential in future exploration missions, 
[revised manuscript text omitted]

Please also note the supplement to this comment:
http://www.geosci-instrum-method-data-syst-discuss.net/gi-2015-25/gi-2015-25-AC1-supplement.pdf
* * *